# Low Mileage, High Fidelity: Evaluating Hypergraph Expansion Methods by Quantifying the Information Loss

## ABSTRACT

Hypergraphs are typically used for solving downstream tasks in two steps: expanding a hypergraph into a conventional graph, known as the *hypergraph expansion*, and conducting machine learning methods on the expanded graph. Depending on how hypergraph expansion is performed, certain information of the original hypergraph may be lost, which negatively affects the accuracy of downstream tasks. If the amount of information loss can be measured, one can select the best hypergraph expansion procedure and target a better downstream performance. To this end, we propose a novel framework, named the MILEAGE, to evaluate hypergraph expansion methods by measuring their degree of information loss. MILEAGE employs the following four steps: (1) expanding a hypergraph; (2) performing the unsupervised representation learning on the expanded graph; (3) reconstructing a hypergraph based on vector representations obtained; and (4) measuring the MILEAGE-score (*i.e.*, mileage) by comparing the reconstructed and the original hypergraphs. To demonstrate the usefulness of MILEAGE, we conduct experiments via downstream tasks on three levels (*i.e.*, node, hyperedge, and hypergraph): node classification, hyperedge prediction, and hypergraph classification on eight real-world hypergraph datasets. We observe that the average and minimum Pearson correlation coefficient between the mileage of expanded graphs and the performance of the downstream task are -0.871 and -0.904, respectively. The results validate that information loss through hypergraph expansion has a negative impact on downstream tasks and MILEAGE can effectively evaluate hypergraph expansion methods through the information loss and recommend a new method that resolves the problems of existing ones.

## CCS CONCEPTS

• **Computing methodologies → Machine learning**.

## KEYWORDS

hypergraph, hypergraph expansion, information loss

**ACM Reference Format:**
Anonymous Author(s). 2018. Low Mileage, High Fidelity: Evaluating Hypergraph Expansion Methods by Quantifying the Information Loss. In *Proceedings of the ACM Web Conference (ACM WWW '24)*. ACM, New York, NY, USA, 10 pages. https://doi.org/XXXXXXX.XXXXXXX

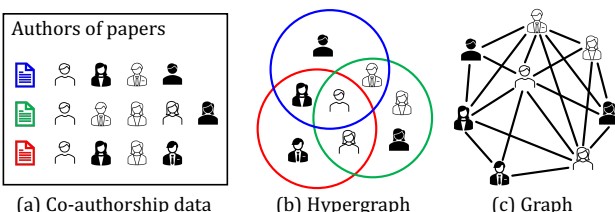

**Figure 1: Co-authorship as a hypergraph and a graph.**

(a) Co-authorship data     (b) Hypergraph     (c) Graph

## 1 INTRODUCTION

*Graphs*, typically defined as a set of nodes connected by edges, are ubiquitous in representing structural information in a variety of domains, ranging from social networks in the Web to molecule structures in biochemistry [20, 29, 45, 48]. However, graphs have limits in representing real-world relationships that are not always *pairwise*. Indeed, in the real-world, there are often *tuplewise relationships* that involve more than two objects [11, 49]. For instance, research papers can be published through the collaboration of more than two co-authors [33]; group communications are increasingly common comparing to 1-on-1 communications in online platforms [3, 39]; and more than two items are often purchased together by users [19] in e-business platforms. Since edges in a conventional graph (a.k.a., graph) encode only *pairwise relationships between two nodes*, they cannot readily represent these tuplewise relationships [11, 47, 49].

A *hypergraph* is a generalization of a graph, consisting of nodes and hyperedges [11, 49]. A *hyperedge* associates an arbitrary number of nodes, and it enables a hypergraph to represent tuplewise relationships [11, 49]. Figure 1 shows an example of representing co-authorship information of three papers (Figure 1-(a)) in a hypergraph (Figure 1-(b)) and in a graph (Figure 1-(c)). In the case of a hypergraph, a set of authors who collaborated on the same paper are grouped in a single hyperedge [3], which is clearly shown as a circle in Figure 1-(b). As a result, in the hypergraph, we can correctly infer who collaborated together on a paper [11, 49]. On the other hand, using a graph, *any possible pair* of authors who collaborated on any paper are connected by an edge [29], as shown in Figure 1-(c). Since an edge indicates a *pairwise* co-authorship, in the graph, we can *only* know whether a pair of authors have collaborated or not (or how frequently they collaborate) but cannot accurately identify who else has collaborated with both of them on a paper.

Thanks to the expressive power of hypergraphs, there have been many attempts to utilize hypergraphs for solving various downstream tasks such as recommendation [18, 46], node classification [7, 12, 44], community detection [9, 38, 39], and hyperedge prediction [21, 47]. Hypergraph mining has often been validated to provide better results, compared to those utilizing graphs. Most of these attempts first expand a hypergraph into a graph, which is named as the *hypergraph expansion* [44], and then conduct machine learning or deep learning on the expanded graphs [8, 12, 18, 21, 39, 44, 46].

The main reason for doing hypergraph expansion is that there are a wide variety of mature algorithms and tools proposed for handling graphs [14, 26, 34, 42] while models particularly designed for hypergraphs are scarse [11, 37]. As a result, while much attention has been put on how to process the extended graph, there has been rare investigation on which hypergraph expansion is more reasonable.

Indeed, there are at least four approaches to hypergraph expansion in literature: (1) *clique expansion* (in short, **C**) [35], (2) *star expansion* (in short, **S**) [2], (3) *multi-level decomposition* (in short, **M**) [11], and (4) *line expansion* (in short, **L**) [44]. In addition, although not explicitly proposed as hypergraph expansion methods, combining the clique and star expansion methods (in short, **CS**) can be a possible hypergraph expansion method.

Downstream tasks are typically built upon one of the above hypergraph expansion methods [7, 8, 12, 18, 21, 39, 44, 46]. However, these studies tend to choose a hypergraph expansion method *without undergoing comparative analysis*, unlike what they do for selecting the machine learning methods for the expanded graphs, which are *carefully evaluated* before deployment. Unfortunately, there is not a clear answer about which hypergraph expansion method is superior and there is also lack of empirical results that systematically compare them. To fill this gap, this paper presents a comprehensive and comparative analysis of hypergraph expansion methods.

We approach this problem by the observation that in hypergraph expansion, every tuplewise relationship is transformed into a set of pairwise relationships. Through this process, certain information encoded by the original hypergraph may have been lost, which may affect the accuracy of downstream tasks negatively. In this paper, we formally define such a problem as the *information loss* in hypergraph expansion. We further categorize the information loss in the hypergraph expansion into the following three types:

- **Non-recoverability:** a problem one cannot recover the original hypergraph precisely from its expanded graph.
- **Tie-weakening:** a problem where the tie strength between nodes belonging to the same hyperedge becomes weaker than that in the expanded graph.
- **Multi-cloning:** a problem where a single node in the original hypergraph is represented by multiple nodes in its expanded graph

When conducting a downstream task, if we can employ a hypergraph expansion method that provides an expanded graph with the *least information loss*, it is expected to provide a higher accuracy in the downstream task. However, to the best of our knowledge, there is *no current way to measure the degree of the information loss* quantitatively. This raises a new research question: how can we quantify the information loss through different hypergraph expansion methods?

To answer this question, we start with an intuition as follows. Given a hypergraph, denoted as $H_o$, and an expanded graph, denoted as $G_1$, expanded via a hypergraph expansion method, it is crucial for $G_1$ to preserve the topology characteristics of $H_o$ as much as possible [44]. *Unsupervised representation learning* (URL) aims to represent nodes in a graph as vectors in a low-dimensional embedding space, where it is essential for these vectors to preserve the underlying topological characteristics of the graph [15, 41]. If we conduct URL on $G_1$ and subsequently reconstruct a hypergraph, denoted as $H_r$, by using the vectors thus obtained, we can quantify the difference between $H_r$ and $H_o$, denoted as $M_1$. Moreover, if we repeat the same process with another expanded graph $G_2$ obtained by a different hypergraph expansion method and quantify the difference $M_2$ in the same way, the difference between $M_1$ and $M_2$ primarily stems from the difference in the two hypergraph expansion methods.

Based on this intuition, in this paper, we propose a novel framework to evaluate hypergraph expansion methods by measuring the information loss in expanded graphs, named the MILEAGE (*Measuring the Information Loss of Expanded grAphs via reconstructinG the hypErgraph*). It consists of the following four steps: (1) expanding a hypergraph into an expanded graph; (2) performing URL on the expanded graph; (3) reconstructing a hypergraph based on the vectors obtained at step (2); and (4) measuring the MILEAGE-score based on the similarity between two sets of hyperedges, one from the reconstructed hypergraph and the other from the original hypergraph.

In this paper, we validate MILEAGE via extensive experiments with eight real-world hypergraphs. If MILEAGE is appropriately designed to evaluate the hypergraph expansion methods through the information loss, we expect to observe a strong *(negative) correlation* between the MILEAGE-score (or mileage thereafter) of the expanded graph and the accuracy of a downstream task conducted on this expanded graph. To validate this claim, we conduct downstream tasks at three-levels (*i.e.*, node, hyperedge, and hypergraph) including node classification, hyperedge prediction, and hypergraph classification. We observe that the average and minimum Pearson correlation coefficient (PCC) [17] between the mileage of the extended graphs and the accuracy of downstream tasks are -0.871 and -0.904 (*i.e.*, fairly high), respectively, across all tasks on all hypergraphs, which indicates MILEAGE is well designed to evaluate the hypergraph expansion methods in terms of the information loss.

We then evaluate the goodness of existing hypergraph expansion methods via MILEAGE, which is interpretable (*i.e.*, can be explained in terms of the information loss), general (*i.e.*, doesn't depend on particular URL methods or datasets), easy to compute, and indicative of the accuracy of downstream tasks. We observe that **CS** generates expanded graphs with the least mileage compared to other hypergraph expansion methods; furthermore, employing **CS** leads to the highest accuracy in all the downstream tasks.

The contributions of this paper are summarized as follows:

- **Problems.** We define three information loss problems (*i.e.*, non-recoverability, tie-weakening, and multi-cloning) appearing in the hypergraph expansion and show that they negatively affect the accuracy of downstream tasks.
- **Novel framework.** We propose a new framework, the MILEAGE, to evaluate hypergraph expansion methods by measuring the degree of the information loss occurring in hypergraph expansion.
- **Extensive evaluation.** Through extensive experiments using eight real-world hypergraph datasets, we first validate the effectiveness of MILEAGE and evaluate the goodness of existing hypergraph expansion methods via MILEAGE. To the best of our knowledge, this is the first paperwork to provide a comprehensive and comparative analysis of hypergraph expansion methods.
- **Recommendation.** Through the information loss analysis and evaluations, we are able to recommend a new and better hypergraph expansion method, a combination of the clique and star expansion (**CS**), which leads to the lowest mileage in expanded graphs and the best performance in downstream tasks.

## 2 RELATED WORK

### 2.1 Hypergraph Expansion Methods

In the literature, four hypergraph expansion methods have been proposed: (1) *clique expansion* [35]; (2) *star expansion* [2]; (3) *multi-level decomposition* [11]; and (4) *line expansion* [44].

Clique expansion (in short, **C**), as shown in Figure 2-(a), expands a hypergraph into a graph [35], where a node corresponds to a node in the hypergraph and an edge does a pair of nodes belonging to the same hyperedge in the hypergraph. As a result, each hyperedge in the hypergraph is represented as a *clique structure* in the graph.

Star expansion (in short, **S**), as shown in Figure 2-(b), expands a hypergraph into a bipartite graph [2]. In a bipartite graph, a node in one side corresponds to a node in the hypergraph, *i.e.*, we refer to this node as an *n-node*, and a node in the other side corresponds to a hyperedge in the hypergraph, *i.e.*, we refer to this node as an *h-node*; an edge between an *n*-node and an *h*-node represents the relationship between a node and its belonging hyperedge in the hypergraph.

Multi-level decomposition (in short, **M**), as shown in Figure 2-(d), expands a hypergraph into *m* (decomposed) graphs [2]. In the level-*i* (decomposed) graph, a node corresponds to a possible combination of *i* nodes belonging to the same hyperedge, and an edge between nodes indicates the two nodes come from the same hyperedge in the hypergraph. The *m* is the largest size of a hyperedge in the hypergraph. The level-1 (decomposed) graph is equivalent to the graph obtained by the clique expansion.

Line expansion (in short, **L**), as shown in Figure 2-(e), expands a hypergraph into a *line graph* [44]. In a line graph, a node (*i.e.*, line node) represents a pair of a node and a hyperedge that the node belongs to and an edge between two line nodes represents (1) the two line nodes have the same hyperedge or (2) they have the same node in the hypergraph.

In addition, although not explicitly proposed in the literature, we can consider a possible hypergraph expansion method by combining **C** and **S** (in short, **CS**). **CS** expands a hypergraph into a heterogeneous graph, as shown in Figure 2-(c), that is a combination of a graph and a bipartite graph obtained by the clique expansion and the star expansion, respectively. In the heterogeneous graph, a node corresponds to a node (*i.e.*, an *n*-node) or a hyperedge (*i.e.*, an *h*-node) in the hypergraph. There are two types of edges: (1) one connects a pair of nodes belonging to the same hyperedge in the hypergraph (*i.e.*, an edge between *n*-nodes) and (2) the other connects the relationship between a node and its belonging hyperedge in the hypergraph (*i.e.*, an edge between an *n*-node and an *h*-node).

### 2.2 Hypergraph Learning for Downstream Tasks

There have been many attempts to propose the methods that utilize hypergraphs in solving downstream tasks such as recommendation [18, 46], node classification [12, 44], community detection [9, 38], and hyperedge prediction [21, 37, 47]. They commonly proceed in two steps: (1) to conduct the hypergraph expansion and (2) to employ machine learning or deep learning methods on the graph obtained by the hypergraph expansion.

These methods were devised after careful consideration on which machine learning (*e.g.*, Louvain [4] or graph cut [10]) or deep learning (*e.g.*, GCN [23] or self-attention [40]) methods to be employed

for a specific downstream task. However, there has been little discussion regarding the selection of hypergraph expansion methods to be used. If a better hypergraph expansion method is chosen after careful analysis of existing hypergraph expansion methods, it is expected that the accuracy of downstream tasks can be further improved. To this end, this paper aims to conduct a comprehensive analysis of hypergraph expansion methods.

## 3 INFORMATION LOSS IN HYPERGRAPH EXPANSION

When a hypergraph is expanded into a graph through the hypergraph expansion, the information of tuplewise relationships that were explicitly represented in the hypergraph may not be fully preserved. This occurs because the hypergraph represents tuplewise relationships by using hyperedges, whereas the graph represents the tuplewise relationships by only using edges, which are *pairwise* in nature. As a result, the explicit representation of tuplewise relationships in the hypergraph may not be fully preserved in the graph, leading to degradation of the downstream task accuracy. In this paper, we define such a problem as *information loss*. In this section, we present the information loss problem associated with each hypergraph expansion method introduced in Section 2.

**Clique expansion (C).** As mentioned in Section 2, **C** represents a hyperedge in a hypergraph as a *clique structure* in a graph, which is the simplest and most intuitive method for representing tuplewise relationships by using pairwise relationships. Due to this simplistic representation, however, **C** cannot fully preserve the precise information in tuplewise relationships (*i.e.*, the hyperedge information in the original hypergraph). This is because every clique structure in the graph can be considered as a hyperedge, regardless of whether it exists due to a hyperedge in the original hypergraph or not.

For example, we see that three nodes $v_a$, $v_b$, and $v_c$ form a hyperedge in Figure 2-(f). However, in the graph obtained by **C** as shown in Figure 2-(a), it is difficult to distinguish whether the three nodes $v_a$, $v_b$, and $v_c$ form a single hyperedge or three pairs of nodes $v_a$ and $v_b$, $v_b$ and $v_c$, and $v_a$ and $v_c$ form separate three hyperedges. Therefore, if we want to recover the original hypergraph from the graph, various hypergraphs, including the original hypergraph, can be considered (Figure 2-(a)). In this paper, we define an information loss problem that can not recover the original hypergraph precisely from its expanded graph as ***non-recoverability***.

To achieve high accuracy in downstream tasks, it is crucial to learn the precise information in tuplewise relationships in a hypergraph [7]. However, the graph obtained by **C** loses some information in hyperedges, making it difficult to learn precise information in the tuplewise relationships. As a result, this information loss could lead to a degradation in the accuracy of downstream tasks.

**Star expansion (S).** Unlike the graph obtained by **C**, the bipartite graph obtained by **S** represents the relationships between nodes and hyperedges, rather than the relationships between nodes. Therefore, the bipartite graph precisely preserves the tuplewise relationships in the hypergraph, indicating that **S** does not suffer from the problem of non-recoverability.

However, we note that there are no edges between nodes of the same side (*i.e.*, edges between *n*-nodes or *h*-nodes) in a bipartite

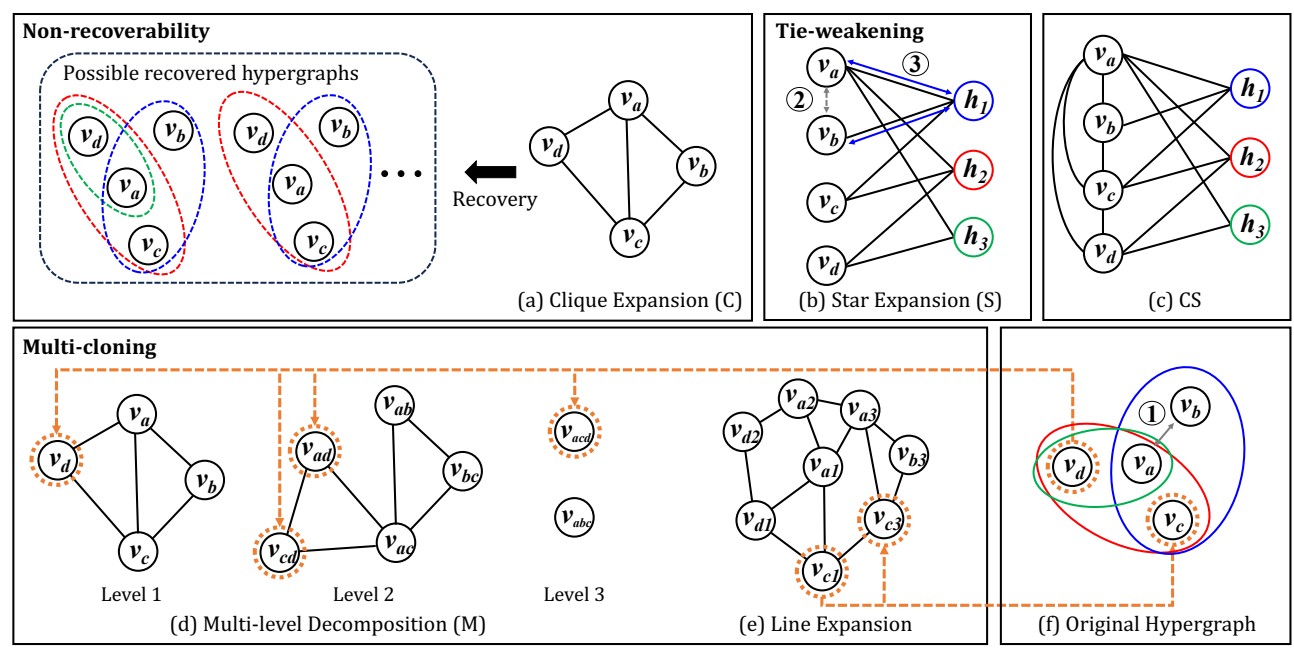

**Figure 2: Three types of information losses in hypergraph expansion methods.**

graph. Therefore, while the original hypergraph represents relationships among nodes *directly*, the bipartite graph represents the relationships among *n*-nodes *indirectly* through *h*-nodes. As a result, there is a problem where the tightness of relationships between nodes in the bipartite graph is weaker than that in the original hypergraph.

For example, in Figure 2-(f)-①, two nodes $v_a$ and $v_b$ have a direct relationship within the blue hyperedge. However, when this hypergraph is expanded into a bipartite graph by **S**, the relationship between the two *n*-nodes needs to be inferred through the *h*-node $h_1$ as shown in Figure 2-(b)-② and Figure 2-(b)-③. In this paper, we define an information loss problem where the tie between nodes in an expanded graph becomes weaker, compared to that in its original graph, as ***tie-weakening***.

In order to provide a higher accuracy in a downstream task, hypergraph-based methods learn relationships between nodes by employing the following intuition [49]: the nodes belonging to the same hyperedge have strong relationships and the nodes belonging to different hyperedges have weak relationships. However, in the bipartite graph, the relationships between *n*-nodes belonging to the same hyperedge are *indirectly* represented through the *h*-node. Consequently, the relationships between *n*-nodes in the bipartite graph are represented weaker, compared to the actual connections in the original hypergraph, thereby likely to affect the accuracy of downstream tasks.

**Multi-level decomposition (M). M** theoretically preserves all tuplewise relationships in the hypergraph by setting the number of levels (*m*) as the largest size of a hyperedge. However, setting *m* in this way requires to manage a large number of graphs, resulting in significant requirements of storage space and computation time. Therefore, *m* is usually set as much smaller than the largest size of hyperedges in practice. For example, in [11], the largest size of hyperedges is 9,705,709 in a real-world hypergraph used in experiments, but *m* is set to only 4 for experimental purposes. In such cases, **M** suffers

from non-recoverability. However, this method does not suffer from the problem of tie-weakening. This is because, as already mentioned in Section 2, the level-1 (decomposed) graph is equivalent to the graph obtained by **C**, and it demonstrates that this decomposed graph represents the direct relationships among the nodes in a hypergraph.

We note that, due to the use of multiple decomposed graphs to preserve hyperedge information precisely, a single node in the hypergraph is represented as multiple nodes in decomposed graphs. For example, in Figure 2-(f), the node $v_d$ is represented in decomposed graphs as follows: in level-1, it is represented as $v_d$ itself; in level-2, it is represented as $v_{ad}$ and $v_{cd}$; and in level-3, it is represented as $v_{acd}$ (Figure 2-(d)). To conduct downstream tasks on decomposed graphs, it is essential to merge the information of multiple nodes in decomposed graphs for representing their corresponding node in the hypergraph. At this time, depending on merging strategies, the merged information may be inappropriate to represent the node in the original hypergraph, which may negatively affect the accuracy of downstream tasks. In this paper, we define such an information loss problem as ***multi-cloning***.

**Line expansion (L).** As mentioned in Section 2, a *line node* in the line graph represents a pair of a hyperedge and a node in an original hypergraph. Therefore, the line graph preserves all tuplewise relationships, which indicates that **L** does not suffer from non-recoverability. Moreover, the line nodes corresponding to the nodes belonging to the same hyperedge are directly connected by edges. This connectivity preserves the tie among the nodes in the same hyperedge; therefore, tie-weakening does not occur in **L**. However, similar to **M**, a single node in the hypergraph can be represented as multiple line nodes in the line graph as shown in Figure 2-(e). Therefore, **L** suffers from the problem of multi-cloning.

**Combining Clique and Star Expansions (CS).** In the case of **CS**, the problem of non-recoverability is addressed by incorporating

 

**Table 1: Summary of information loss problems associated with hypergraph expansion methods**

|  | C | S | M | L | CS |
|---|---|---|---|---|---|
| Non-recoverability | ✗ |  | ✗ |  |  |
| Tie-weakening |  | ✗ |  |  |  |
| Multi-cloning |  |  | ✗ | ✗ |  |

**S**, and the problem of tie-weakening is addressed by incorporating **C**. Therefore, this method is free from both problems of non-recoverability and tightness-weakening. In addition, since the nodes in the hypergraph are not represented by multiple nodes in the expanded graph, there is no multi-cloning as well.

Here, we provide a summary of the three information loss problems associated with each expansion method in Table 1.

## 4 MILEAGE: THE PROPOSED FRAMEWORK

If we employ a hypergraph expansion method that provides an expanded graph with a smaller information loss in conducting a downstream task, it can be expected to provide more-accurate results. Unfortunately, there is no such work that aims to measure the degree of the information loss of expanded graphs. This motivates us to propose a framework, named the MILEAGE, to evaluate the hypergraph expansion methods.

**Overview.** MILEAGE evaluates hypergraph expansion methods by measuring the mileage (*i.e.*, the degree of the information loss) based on the following intuition. A graph (say, $G_1$) expanded by a hypergraph expansion method from a hypergraph (say, $H_o$) should preserve the topology of $H_o$ as much as possible [44]. *Unsupervised representation learning* (URL) aims to represent nodes in a graph as vectors in a low-dimensional embedding space by leveraging the topology of the graph; it is essential for these vectors to preserve the underlying topological properties of the graph [15, 41]. If we conduct URL on $G_1$ and reconstruct a hypergraph (say, $H_{r_1}$) based on the vectors obtained by URL, we can quantify the mileage ($M_1$) of $G_1$ by simply measuring the difference between $H_{r_1}$ and $H_o$.

We note that the difference between $H_{r_1}$ and $H_o$ can be attributed not only to the information loss obtained through the *hypergraph expansion* but also to other factors, such as *URL* methods and *hypergraph reconstruction* methods. Thus, it is difficult to say that the information loss solely contributes to $M_1$. However, if we repeat the same process with another expanded graph $G_2$ obtained by a different hypergraph expansion method and measure the information loss $M_2$ of $G_2$ in the same way, the difference between $M_1$ and $M_2$ primarily stems from the difference in the two hypergraph expansion methods employed. Therefore, we can evaluate hypergraph expansion methods *by quantifying* the information loss.

Figure 3 illustrates an example of the overall process of MILEAGE to quantify the information loss of an expanded graph obtained by **S** from a given hypergraph based on the above intuition. It consists of four components: (1) *hypergraph expansion*, (2) *node representation*, (3) *hypergraph reconstruction*, and (4) *MILEAGE-score computation*. We describe each component in detail in the rest of this section and provide the time complexity analysis of MILEAGE in Appendix A.1.

**Hypergraph expansion.** *Hypergraph expansion* (Figure 3-(a)) is a component that transforms a given hypergraph into an expanded

graph, selecting one of the five hypergraph expansion methods (*i.e.*, **C**, **S**, **M**, **L**, or **CS**) mentioned in Section 2.

**Node representation.** *Node representation* (Figure 3-(b)) is a component that represents nodes of the expanded graph into vectors in a low-dimensional space by leveraging the topological characteristics of the graph. We note that *our MILEAGE is agnostic to URL methods*; thus, any URL method can be employed. In this paper, we employ DGI [41], which is a widely used URL method and provides the best result in our case.[1]

**Hypergraph reconstruction.** *Hypergraph reconstruction* (Figure 3-(c)) is a component that reconstructs a hypergraph based on the vectors obtained from node representation. In this paper, we conduct hypergraph reconstruction in the same manner as done in the previous work [37, 47], based on the vectors obtained from node representation. This approach first conducts hyperedge prediction to determine a set of hyperedges whose number is equal to the number of the original hyperedges. Then, it reconstructs a hypergraph where the predicted hyperedges are considered as its hyperedges.

To facilitate hyperedge prediction in a practical sense, it is necessary to have a set of candidate hyperedges for determining the final hyperedges [21, 37, 47]. To this end, we begin by constructing a set of candidate hyperedges composed of existing hyperedges (*i.e.*, positive hyperedges) and non-existing hyperedges (*i.e.*, negative hyperedges). To generate negative hyperedges, we employ *Clique Negative Sampling* [21, 31][1], which is commonly used in negative sampling, and generate negative hyperedges whose number is equal to the number ($h$) of positive hyperedges. This sampling method selects a random hyperedge from positive hyperedges and replaces a random constituent node in the hyperedge with a random node that is adjacent to all other constituent nodes but belongs to different hyperedges.

Afterward, to determine $h$ hyperedges among the $2h$ candidate hyperedges that are most likely to be true positive hyperedges, we measure the degree of positiveness for all $2h$ hyperedges by using the prediction approach below. Based on the measured scores, we choose the top-$h$ hyperedges as positive hyperedges and regard them as actual hyperedges in the reconstructed hypergraph. To achieve this, we employ two prediction approaches: (1) *heuristic-based prediction* and (2) *model-based prediction*. Both approaches rely on the intuition that the vectors of nodes belonging to the same hyperedge should be close to each other (*i.e.*, similar) in the embedding space, while the vectors of the nodes belonging to different hyperedges should be distant (*i.e.*, dissimilar) in the embedding space [21, 37, 47]. The heuristic-based prediction, which does not require a separate training process, measures the degree of positiveness by computing the average similarity between all pairs of nodes belonging to the same hyperedge. In this paper, we use the dot product as a similarity measure: *i.e.*, the similarity between two nodes is measured by taking the dot product of their respective vectors. In contrast to the heuristic-based prediction, the model-based prediction first trains a prediction model by using positive and negative hyperedges and determines the degree of positiveness of a given hyperedge via the prediction model [21, 37, 47]. In this paper, we simply use multilayer perceptron (MLP) [13] as a prediction model.

---

[1]The issues of selecting URLs and negative sampling methods are described in detail in EQ2 and EQ4 in Section 5.2, respectively.

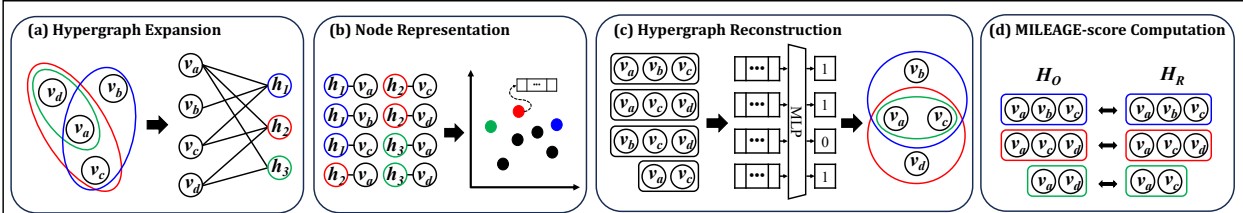

**Figure 3: Overview of the MILEAGE.**

**MILEAGE-score computation.** *MILEAGE-score computation* (Figure 3-(d)) is a component that quantifies the information loss of the expanded graph by comparing the reconstructed and original hypergraphs. In this paper, we regard the degree of mismatch between the hyperedges in the reconstructed hypergraph and those of the original hypergraph as the mileage; we measure the degree of mismatch based on *precision* [17]. Given the original hypergraph $H_o = (V, HE_o)$ and the reconstructed hypergraph $H_r = (V, HE_r)$, where $V$ indicates a set of nodes and $HE_o$ and $HE_r$ indicate sets of hyperedges of $H_o$ and $H_r$, respectively, the mileage is computed as shown in the following equation:

$$mileage\left(H_o, H_r\right) = 1 - precision$$
$$= 1 - \frac{|HE_o \cap HE_r|}{|HE_r|}. \qquad (1)$$

A smaller mileage of the expanded graph indicates that the reconstructed hypergraph from the expanded graph is more similar to the original hypergraph. This suggests that the expanded graph has a smaller information loss.

## 5 EVALUATION

In this section, we first validate the usefulness of the MILEAGE and evaluate the goodness of existing hypergraph expansion methods via extensive experiments. Our experiments are designed by aiming to answer the following evaluation questions (EQs):

- **EQ1**: Are the *information loss* of the expanded graph and the accuracy of downstream tasks correlated?
- **EQ2**: Is MILEAGE agnostic to URL methods?
- **EQ3**: Which hypergraph expansion method provides the least information loss in its expanded graph?
- **EQ4**: Is MILEAGE agnostic to negative sampling methods?
- **EQ5**: How do different values of parameters of MILEAGE influence the correlation between the information loss and the accuracy of downstream tasks?

### 5.1 Experimental Settings

We conduct the experiments on a server equipped with Intel i9-9900K, 500GB SSD, 64GB memory, GeForce RTX 2070, and Linux 5.4.0-53. For reproducibility, we provide our detailed settings such as parameters of MILEAGE in Appendix A.2.

**Datasets.** We use eight datasets of real-world hypergraph datasets prevalent in four distinct domains (*i.e.*, graph labels): Senate [9], House [9], Primary [9], High [9], Enron [3, 16], Eu [3], Substances [3], and Classes [3]. We preprocess each hypergraph to remove the hyperedges with a size of 1 (*i.e.*, the number of nodes in a hyperedge is 1) and duplicate hyperedges, following [25]. We summarize their basic statistics of the datasets in Table 2 and provide their detailed descriptions in Appendix A.3.

**Table 2: Summary of the datasets used in the experiments**

| Dataset | Domain | # of Nodes | # of Hyperedges | # of Labels |
|---|---|---|---|---|
| **Senate** **House** | Committee | 282 1,290 | 302 335 | 2 2 |
| **Primary** **High** | Contact | 242 327 | 12,704 7,818 | 11 9 |
| **Enron** **Eu** | Email | 143 946 | 1,459 24,520 | - - |
| **Substances** **Classes** | NDC | 3,767 1,149 | 6,631 1,049 | - - |

**Target downstream tasks.** In order to demonstrate the usefulness of MILEAGE regardless of downstream tasks, we conduct downstream tasks at three-levels (*i.e.*, node, hyperedge, and hypergraph) such as node classification, hyperedge prediction, and hypergraph classification. The procedure for each downstream task is as follows. First, we construct 10 different training-test splits for each dataset. At this time, the ratio of training to test is set to 8:2. Then, for each split, we (1) train a classifier model; (2) predict the class labels (*i.e.*, labels of nodes, positive/negative of hyperedges, and domains of hypergraphs) for the test set via the model; and (3) measure the accuracy of the prediction results. Finally, the final accuracy is computed by averaging the accuracies of all splits. In order to show consistent results, for each downstream task, we use four well-known classifier models, *i.e.*, logistic regression (in short, LR) [24], random forest (in short, RF) [5], support vector machine (in short, SVM) [30], and multilayer perceptron (in short, MLP) [13].

### 5.2 Experimental Results

**EQ1 and EQ2: Correlation between the information loss and the downstream task accuracy.** If the information loss actually affects the accuracy of downstream tasks, the accuracy is expected to decrease as the information loss increases. Moreover, if MILEAGE is properly designed to evaluate the hypergraph expansion methods through the information loss, there should appear a clear *(negative) correlation* between the mileage of expanded graphs and the accuracy of downstream tasks conducted on these graphs.

To investigate this claim, for each dataset, we first measure the mileage of each expanded graph via MILEAGE and then conduct the three downstream tasks on the expanded graph with measuring the accuracy. Next, we rank the expansion methods in the mileage and the accuracy in descending order and compute the Pearson correlation coefficient (PCC) [17] between the rankings in the mileage and the accuracy of the downstream task. At this time, to demonstrate the *agnostic nature* of MILEAGE to URL methods, we use four URL methods: (1) Deepwalk [32], (2) Node2vec [15], (3) DGI [39], and (4) BGRL [36].

**Table 3: PCC between rankings in information loss obtained by MILEAGE and downstream task accuracy**

| | | Node Classification | | | | Hyperedge Prediction | | | | Hypergraph Classification | | | | Avg. |
|---|---|---|---|---|---|---|---|---|---|---|---|---|---|---|
| | | LR | RF | SVM | MLP | LR | RF | SVM | MLP | LR | RF | SVM | MLP | |
| **Model** | Deepwalk | -0.798 | -0.821 | -0.869 | -0.861 | -0.804 | -0.889 | -0.798 | -0.869 | -0.905 | -0.885 | -0.824 | -0.865 | -0.849 |
| | Node2vec | -0.865 | -0.854 | -0.881 | -0.905 | -0.932 | -0.801 | -0.915 | -0.903 | -0.911 | -0.896 | -0.843 | -0.854 | -0.880 |
| | DGI | -0.820 | -0.886 | -0.923 | -0.951 | -0.865 | -0.919 | -0.826 | -0.968 | -0.913 | -0.923 | -0.884 | -0.971 | -0.904 |
| | BGRL | -0.818 | -0.864 | -0.902 | -0.896 | -0.845 | -0.901 | -0.809 | -0.946 | -0.912 | -0.901 | -0.885 | -0.890 | -0.882 |
| | Average | -0.825 | -0.856 | -0.893 | -0.903 | -0.862 | -0.881 | -0.837 | -0.921 | -0.910 | -0.901 | -0.859 | -0.895 | -0.879 |
| **Heuristic** | Deepwalk | -0.802 | -0.815 | -0.861 | -0.856 | -0.805 | -0.887 | -0.785 | -0.862 | -0.902 | -0.883 | -0.821 | -0.871 | -0.845 |
| | Node2vec | -0.835 | -0.880 | -0.885 | -0.753 | -0.875 | -0.817 | -0.800 | -0.922 | -0.841 | -0.863 | -0.872 | -0.891 | -0.853 |
| | DGI | -0.819 | -0.881 | -0.918 | -0.910 | -0.913 | -0.934 | -0.902 | -0.894 | -0.873 | -0.832 | -0.855 | -0.912 | -0.895 |
| | BGRL | -0.812 | -0.866 | -0.894 | -0.887 | -0.831 | -0.897 | -0.795 | -0.934 | -0.893 | -0.817 | -0.807 | -0.881 | -0.860 |
| | Average | -0.817 | -0.860 | -0.890 | -0.852 | -0.856 | -0.883 | -0.821 | -0.903 | -0.877 | -0.874 | -0.839 | -0.889 | -0.863 |
| **Overall average** | | -0.821 | -0.858 | -0.892 | -0.876 | -0.859 | -0.882 | -0.829 | -0.911 | -0.894 | -0.888 | -0.849 | -0.892 | -0.871 |

In the cases of **L** and **M**, as already mentioned in Section 3, these methods expand a single node in a hypergraph into multiple nodes in their expanded graph. Therefore, it is essential to aggregate the vectors of multiple nodes in decomposed graphs for the vector of its corresponding node in the hypergraph to conduct downstream tasks. To conduct hyperedge prediction, in the case of **L**, following [44], we average the vectors of multiple nodes as a vector of a corresponding node in the hypergraph. Similarly, in the case of **M**, we average the vectors of multiple nodes as a vector of a corresponding node for each level, taking the averaged vectors in *all* levels for the final vector of the corresponding node.

In the case of node classification, we note that only four hypergraphs have node labels among all the hypergraphs. Therefore, we conduct a node classification task on these four hypergraphs only. In the case of hypergraph classification, since we have only eight hypergraphs, the amount of data for model training is insufficient. To address this issue, we sample 100 sub-hypergraphs whose sizes are 10%, 15%, and 20% of the total nodes from each hypergraph via *forest fire* [28]. Furthermore, to conduct hypergraph classification, we need a vector for a hypergraph. Since we only have vectors of nodes, following [43], we compute a vector of each graph by averaging of the vectors of all nodes in the hypergraph.

Table 3 shows PCC between the mileage and the accuracy of a downstream task. The rows correspond to two variants of MILEAGE in terms of hyperedge prediction methods (*i.e.*, model-based and heuristic-based) in hypergraph reconstruction. A row is subdivided into four categories, indicating the URL methods used for node representation. The columns correspond to downstream tasks. A column is subdivided into four categories, indicating the classifiers used for a downstream task.

We summarize the results shown in Tables 3 as follows. First, we observe that there is a *strong (negative) correlation* between the mileage and the downstream task accuracy in all cases regardless of variants of MILEAGE, classifiers, and downstream tasks. More specifically, the average and minimum PCCs are -0.871 and -0.904, respectively, which are very high. Second, we observe strong (negative) correlations in all URL methods; DGI exhibits the strongest (negative) correlation among the four URL methods. Third, in all URL methods, we observe that the model-based prediction shows a slightly stronger correlation than the heuristic-based one.

Through the results, we have validated that (1) the information loss problem we have defined has a truly negative impact on the accuracy of downstream tasks, (2) MILEAGE is *agnostic* to URL methods, and (3) MILEAGE appropriately evaluates the hypergraph expansion methods. Moreover, we have confirmed that MILEAGE using model-based prediction with DGI provides the strongest (negative) correlation among all possible variants of MILEAGE; therefore, we use it as our MILEAGE in the subsequent experiments.

**EQ3: Comparison of hypergraph expansion methods.** Next, we evaluate the goodness of existing hypergraph expansion methods in terms of the information loss via MILEAGE. Tables 4 and 5 show the mileage of expanded graphs obtained via MILEAGE and the accuracy in a downstream task using DGI as a URL method and MLP as a classifier on expanded graphs, respectively.

We summarize the results shown in Tables 4 and 5 as follows. First, **M** provides expanded graphs that provide the *highest* mileage (*i.e.*, highest information loss). Among hypergraph expansion methods, **M** is the only one that has two kinds of information loss problems (*i.e.*, non-recoverability and multi-cloning). Therefore, it appears to have the lowest performance in terms of reconstructing the hypergraph. Second, **CS** provides expanded graphs with the *least* mileage. We conjecture that the mileage of **CS** primarily stems from in URL and hypergraph reconstruction, as **CS** does not exhibit any information loss problems during the hypergraph expansion process. Third, we observe that the trend of downstream task accuracy is opposite to that of the mileage. This demonstrates once again that there is a clear (negative) correlation between the mileage and the accuracy.

Through the results, we conclude that (1) **CS** provides the best expanded graphs in terms of a mileage and (2) it is possible to estimate the performance superiority of the downstream task on expanded graphs, based on the superiority of the mileage obtained by MILEAGE.

**EQ4: Sensitivity of MILEAGE according to negative sampling methods in hypergraph reconstruction.** Next, we analyze the change of the mileage according to different negative sampling methods in MILEAGE. For negative sampling methods, we employ the following three methods [21, 31]: (1) *Sized Negative Sampling* (SNS), (2) *Motif Negative Sampling* (MNS), and (3) *Clique Negative Sampling*. Figure 4 shows the mileage of expanded graphs in (a)

**Table 4: Mileage and (ranking) of expanded graphs obtained by our framework (lower is better)**

| Dataset | C | S | M | L | CS |
|---|---|---|---|---|---|
| Senate | 0.699 (1) | 0.430 (4) | 0.532 (2) | 0.458 (3) | 0.411 (5) |
| House | 0.574 (2) | 0.537 (3) | 0.581 (1) | 0.534 (4) | 0.419 (5) |
| Primary | 0.509 (3) | 0.419 (5) | 0.513 (2) | 0.620 (1) | 0.472 (4) |
| High | 0.495 (2) | 0.400 (4) | 0.524 (1) | 0.485 (3) | 0.345 (5) |
| Enron | 0.229 (4) | 0.300 (3) | 0.452 (2) | 0.485 (1) | 0.157 (5) |
| Eu | 0.375 (5) | 0.432 (2) | 0.422 (3) | 0.498 (1) | 0.418 (4) |
| Substances | 0.675 (1) | 0.582 (4) | 0.637 (2) | 0.625 (3) | 0.525 (5) |
| Classes | 0.863 (2) | 0.809 (4) | 0.873 (1) | 0.856 (3) | 0.750 (5) |
| **Avg. Ranking** | 2.5 | 3.3 | 1.8 | 2.4 | 4.4 |

**Table 5: Accuracy and (ranking) of downstream tasks on expanded graphs (higher is better)**

| | Dataset | C | S | M | L | CS |
|---|---|---|---|---|---|---|
| **Node Classification** | Senate | 0.419 (3) | 0.435 (2) | 0.412 (5) | 0.415 (4) | 0.440 (1) |
| | House | 0.481 (3) | 0.484 (2) | 0.463 (5) | 0.476 (4) | 0.486 (1) |
| | Primary | 0.487 (3) | 0.498 (1) | 0.477 (4) | 0.476 (5) | 0.497 (2) |
| | High | 0.457 (4) | 0.471 (2) | 0.453 (5) | 0.461 (3) | 0.473 (1) |
| | Avg. Ranking | 4.3 | 1.8 | 4.8 | 4.0 | 1.3 |
| **Hyperedge Prediction** | Senate | 0.492 (3) | 0.498 (2) | 0.481 (5) | 0.492 (3) | 0.505 (1) |
| | House | 0.500 (2) | 0.499 (3) | 0.485 (5) | 0.490 (4) | 0.508 (1) |
| | Primary | 0.500 (3) | 0.502 (2) | 0.488 (5) | 0.492 (4) | 0.539 (1) |
| | High | 0.499 (3) | 0.501 (2) | 0.463 (5) | 0.482 (4) | 0.502 (1) |
| | Enron | 0.498 (4) | 0.501 (2) | 0.498 (4) | 0.499 (3) | 0.519 (1) |
| | Eu | 0.562 (1) | 0.553 (2) | 0.500 (4) | 0.499 (5) | 0.535 (3) |
| | Substances | 0.491 (5) | 0.512 (2) | 0.503 (4) | 0.510 (3) | 0.521 (1) |
| | Classes | 0.496 (2) | 0.489 (4) | 0.485 (5) | 0.495 (3) | 0.513 (1) |
| | Avg. Ranking | 2.9 | 2.4 | 4.6 | 3.6 | 1.3 |
| **Hypergraph Classification** | S10% | 0.338 (4) | 0.432 (2) | 0.304 (5) | 0.431 (3) | 0.466 (1) |
| | S15% | 0.416 (4) | 0.463 (2) | 0.401 (5) | 0.457 (3) | 0.488 (1) |
| | S20% | 0.463 (3) | 0.487 (2) | 0.424 (5) | 0.459 (4) | 0.492 (1) |
| | Avg. Ranking | 4.7 | 2.0 | 6.0 | 4.0 | 1.0 |
| **Overall Avg. Rank.** | | 3.7 | 2.0 | 5.0 | 3.3 | 1.0 |

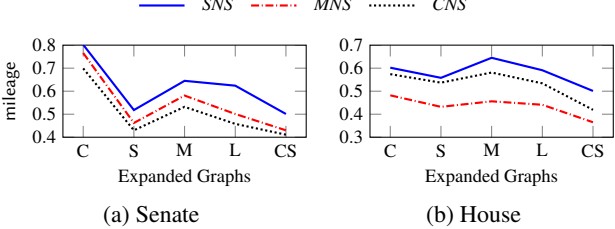

**Figure 4: mileage change according to negative sampling methods.**

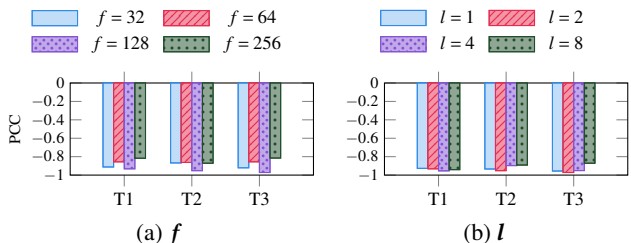

**Figure 5: PCC change according to parameters in MILEAGE.**

Next, we analyze the change of the mileage by different numbers of layers ($l$). Following [23, 40], we set the number of layers in MILEAGE as 1, 2, 4, and 8. Figure 5-(b) shows the results. The $x$-axis represents the number of layers in MILEAGE for each downstream task, and the $y$-axis does the mileage. The mileage shows two trends: (1) the score steadily decreases up to a certain number of layers ($l$=2); and (2) after the point, as the number of layers increases, the score gradually increases. Overall, this change of the mileage is not significant, which indicates our framework is *insensitive* to $l$ as well.

## 6 CONCLUSION AND FUTURE WORK

In this paper, we point out three information loss problems occurring in the hypergraph expansion, *i.e.*, non-recoverability, tie-weakening, and multi-cloning. We then propose a novel framework, the MILEAGE, that evaluates the hypergraph expansion methods through the information loss in four steps: (1) expanding a hypergraph into an expanded graph; (2) performing the URL on the expanded graph; (3) reconstructing a hypergraph based on the vectors thus obtained; and (4) quantifying the information loss by the mileage that indicates the difference between the reconstructed and original hypergraphs. We validate the usefulness of MILEAGE via extensive experiments with three downstream tasks on eight real-world hypergraphs. Through the results, we observe that (1) the information loss problems we have defined incur a truly negative impact on downstream tasks and (2) MILEAGE is well designed to evaluate hypergraph expansion methods. Furthermore, we evaluate existing hypergraph expansion methods via MILEAGE. We observe that a new combination of Clique and Star Expansions (**CS**) produces expanded graphs with the least mileage, even though it is not covered in the literature. Our work inspires the development of novel hypergraph expansion methods and hypergraph-based machine learning methods for solving downstream tasks. As future work, we plan to study (a) hypergraph reconstruction methods using generative artificial intelligence such as large-language models and (b) hypergraph-based methods for downstream tasks (*e.g.*, recommendation and hypergraph classification) using the **CS** expansion.

Senate and (b) House. The $x$-axis represents expanded graphs and the $y$-axis does the mileage. We observe that even with different negative sampling methods, the trend of the mileage remains nearly consistent, which indicates our framework is *insensitive* to negative sampling methods.

**EQ5: Parameter sensitivity of MILEAGE.** It is desirable that the correlation between the mileage and the downstream task accuracy is not changed with different values of parameters of MILEAGE. To investigate this, we first analyze the change of PCC according to different dimensionalities ($f$) in MILEAGE. Following [22, 27], we set dimensionalities of MILEAGE as 32, 64, 128, and 256. Figure 5-(a) shows the results. The $x$-axis represents dimensionalities in MILEAGE for each downstream task where T1 represents node classification; T2 represents hyperedge prediction; and T3 represents hypergraph classification. The $y$-axis does PCC. We observe that even in different dimensionalities, PCC remains nearly consistent, which indicates MILEAGE is *insensitive* to $f$ as well.

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

# A APPENDIX

In this appendix, we describe (1) time complexity of MILEAGE (Appendix A.1), (2) the detailed experimental settings of our experiments (Appendix A.2), and (3) detailed descriptions of datasets (Appendix A.3).

## A.1 Time Complexity of MILEAGE

In this section, we analyze the time complexity for one epoch of MILEAGE as follows. For the sake of simplicity, we assume MILEAGE to use Node2vec for node representation and MLP with $l$ layers, the dimensionality of $f$, and $m$-level multi-level decomposed graphs for hypergraph reconstruction. Given a hypergraph with $v$ nodes (average degree is $d$) and $h$ hyperedges (average size is $k$), the time complexity of each component is as follows: the time complexity of hypergraph expansion to get five expanded graphs becomes $O(k^2+hk+\sum_{i=0}^{k}|h|\binom{k}{i}^2+hk^2\ nd^2+hk^2+hk)$; the time complexities of node representation and hypergraph reconstruction are $O(n^2)$ and $O(2h^2fl^2+2hl^2)$ [6, 14], respectively; and the time complexity of mileage-score computation becomes $O(h^2)$. Therefore, the overall time complexity of MILEAGE becomes $O(n^2+h^2)$, since $k$, $d$, $f$, and $l$ are much smaller than $v$ or $h$ and thus treated as constants.

## A.2 Reproducibility

In this section, we describe the detailed settings of our experiments. The experiments are conducted on a server equipped with Intel i9-9900K, 500GB SSD, 64GB memory, GeForce RTX 2070, and Linux 5.4.0-53. For URL methods, we use the source codes provided by the authors [15, 32, 36, 41]. For parameters for each method, we use the best setting found via extensive grid search in the ranges suggested in its respective paper. We carefully tuned the hyperparameters of our

Mileage. We set the feature dimensionality ($f$) as 128, the number of epochs as 200, the activation function as ReLU [1], the number of layers ($l$) as 2, and the learning rate as 0.001 for all the datasets. For the level ($m$) of M, we set 4 for Enron, 3 for Senate, Primary, High, EU, and Classes, 2 for House and Substances.

## A.3 Datasets

In this section, we provide detailed descriptions of datasets.

- **Senate and House:** Senate and House are datasets that represent the committee memberships in the US Senate and the US House of Representatives, respectively. A node corresponds to a senator (or a representative) and a hyperedge indicates a set of senators (or representatives) of a committee membership. The class label of a node indicates a political affiliation.
- **Primary and High:** Primary and High are datasets that represent the interactions of students in classrooms in the primary school and the high school, respectively. A node corresponds to a student and a hyperedge indicates a set of students interacting each other as a group during an unit interval. The class label of a node indicates a classroom where the student belongs to.
- **Enron and EU:** Enron and EU are datasets that represent the email communications in Enron and European research institution, respectively. A node corresponds to an email address and a hyperedge indicates a set of email addresses of the sender and all recipients.
- **Substances:** Substances is a dataset that represents the substances and drugs. A node corresponds to a substance and a hyperedge indicates a set of substances of a drug.
- **Classes:** Classes is a datasets that represents the drugs and class labels applied to drugs. A node corresponds to a class label and a hyperedge indicates a set of class labels of a drug.

