# OpenReview forum: "Low Mileage, High Fidelity: Evaluating Hypergraph Expansion Methods by Quantifying the Information Loss"
_ACM.org/TheWebConf/2024/Conference — TheWebConf24 Oral_

### Official Review · Reviewer_BfWy · 2023-11-21

**Novelty:** 6
**Technical Quality:** 6

**Review:**

### Quality
The paper introduces MILEAGE, a framework for evaluating hypergraph expansion methods by measuring the degree of information loss. The quality of research is high, with a clear problem statement, a proposed solution, and a robust evaluation method. The experimental setup is comprehensive, covering eight real-world datasets.

### Clarity
The paper is well-written, with a clear structure and a logical flow from the problem introduction through to the proposed solution and empirical evaluation. Key concepts are explained with sufficient background.

### Originality
This work appears to be original, filling a gap in the evaluation of hypergraph expansion methods. The idea of quantifying information loss to assess the fidelity of hypergraph expansion is innovative and could be a significant contribution to the field.

### Significance
The significance of this research is high, given the importance of accurate hypergraph expansion in various applications. The ability to measure and minimize information loss during expansion could improve the performance of downstream tasks such as node classification and hyperedge prediction.

### Pros

* MILEAGE is a new approach to evaluate hypergraph expansion methods.
* The paper performs thorough experiments on eight real-word datasets to evaluate the proposed method.

### Cons

* While the paper does provide a comparative analysis, additional comparison with other hypergraph expansion evaluation methods could be beneficial.

**Questions:**

N/A

**Reviewer Confidence:**

4: The reviewer is certain that the evaluation is correct and very familiar with the relevant literature

**Scope:**

4: The work is relevant to the Web and to the track, and is of broad interest to the community

---

### Official Review · Reviewer_5cKf · 2023-11-22

**Novelty:** 6
**Technical Quality:** 6

**Review:**

The authors present a method for evaluating hypergraph expansion techniques, i.e. converting hypergraphs into graphs. Their approach involves a vector representation of the expanded graph, used to reconstruct the original hypergraph. Then they measure the dissimilarity between the reconstructed and original hypergraphs to quantify the "information loss" and hence assess the expansion quality.

The paper is well-composed with well-substantiated claims and provides a thorough experimental analysis, yielding convincing results.

Suggestions for Improvement:

- The authors should elaborate on their method for reconstructing the hypergraph, as this is a critical component of their framework. Simply referring to other works detracts from the paper’s self-sufficiency.

- Currently, the authors employ only traditional machine learning methods for downstream tasks. Incorporating and analyzing the performance with contemporary deep learning models could provide additional insights.

- In Tables 4 and 5, rank the results to start with the best (instead of the highest) might more clearly highlight that Mileage is a reliable predictor of Accuracy.

**Questions:**

Which impact would have the use of deep learning models for downstream tasks?

**Ethics Review Description:**

No isssues found

**Reviewer Confidence:**

3: The reviewer is confident but not certain that the evaluation is correct

**Scope:**

3: The work is somewhat relevant to the Web and to the track, and is of narrow interest to a sub-community

---

### Official Review · Reviewer_joqJ · 2023-11-26

**Novelty:** 4
**Technical Quality:** 5

**Review:**

Pros:

Defines three types of information loss problems in hypergraph expansion methods: non-recoverability, tie-weakening, and multi-cloning.

Proposes a novel framework (MILEAGE) to quantify information loss from hypergraph expansion by reconstructing the hypergraph and comparing to the original. Shows strong negative correlation between mileage score and downstream accuracy.
Evaluates existing expansion methods and finds combining clique and star expansion gives best mileage score and downstream accuracy.

Cons:

The MILEAGE framework relies on being able to accurately reconstruct the original hypergraph, which may be difficult in some cases. More analysis could be done on when the framework breaks down.

**Questions:**

How sensitive is the mileage score to changes in the hypergraph reconstruction process, like changes to the URL method or negative sampling?

For very large hypergraphs, the reconstruction process could get expensive. Are there ways to approximate or speed up parts of MILEAGE to make it more scalable?

**Reviewer Confidence:**

3: The reviewer is confident but not certain that the evaluation is correct

**Scope:**

4: The work is relevant to the Web and to the track, and is of broad interest to the community

---

### Official Review · Reviewer_8byM · 2023-11-27

**Novelty:** 5
**Technical Quality:** 4

**Review:**

### Summary

The paper introduces a novel framework named MILEAGE, designed to evaluate hypergraph expansion methods by quantifying their information loss. Hypergraphs are often expanded into conventional graphs, a process during which some original information may be lost, potentially affecting the accuracy of downstream tasks. The paper summarizes three types of situations that may lead to information loss (1) Non-recoverability, (2) Tie-weakening,  (3)Multi-cloning.

MILEAGE addresses this by following four steps: (1) expanding a hypergraph, (2) performing unsupervised representation learning on the expanded graph, (3) reconstructing a hypergraph based on the obtained vector representations, and (4) measuring the MILEAGE-score by comparing the reconstructed and original hypergraphs. This framework is tested on various real-world hypergraph datasets across different tasks, including node classification, hyperedge prediction, and hypergraph classification. The study finds a strong negative correlation between the information loss in expanded graphs (MILEAGE) and the performance of downstream tasks, demonstrating the effectiveness of MILEAGE in evaluating and improving hypergraph expansion methods.

### Strength

- The MILEAGE framework introduces a novel way to evaluate hypergraph expansion methods.
- The framework provides a measurable and quantifiable way to assess the degree of information loss.
- The paper is well written and organized.

### Weakness

- The paper lacks a discussion on the advantages of the MILEAGE framework. Since it shows a strong correlation with the performance of downstream tasks, why not directly use the performance to evaluate the quality of a method?
- MILEAGE is limited to assessing the overall loss of three type of information loss. Is there a more detailed method available to evaluate the extent of each specific type of information loss?

**Questions:**

Please refer to the weaknesses.

**Reviewer Confidence:**

2: The reviewer is willing to defend the evaluation, but it is likely that the reviewer did not understand parts of the paper

**Scope:**

3: The work is somewhat relevant to the Web and to the track, and is of narrow interest to a sub-community

---

### Decision · Program_Chairs · 2024-01-22

**Decision:**

Accept (Oral)

**Comment:**

The reviewers and the area chair agree that this is a well written paper that studies an interesting problem and has nontrivial results. We recommend acceptance.